# The STress-And-Coping suppoRT Intervention (START) for Chinese Women Undergoing Abortion: A Randomized Controlled Trial Protocol

**DOI:** 10.3390/ijerph19116611

**Published:** 2022-05-28

**Authors:** Na Wang, Xiu Zhu, Jenny Gamble, Elizabeth Elder, Jyai Allen, Debra K. Creedy

**Affiliations:** 1School of Nursing and Midwifery, Gold Coast Campus, Griffith University, Parklands Drive, Gold Coast, QLD 4215, Australia; na_wang_3@163.com (N.W.); e.elder@griffith.edu.au (E.E.); 2School of Nursing, Capital Medical University, 10 Xitoutiao Road, Fengtai District, Beijing 100069, China; 3School of Nursing, Peking University, 38 Xueyuan Road, Haidian District, Beijing 100191, China; 4School of Nursing and Midwifery, Logan Campus, Griffith University, University Drive, Meadowbrook, Brisbane, QLD 4131, Australia; j.gamble@griffith.edu.au (J.G.); jyai.allen@griffith.edu.au (J.A.); d.creedy@griffith.edu.au (D.K.C.)

**Keywords:** abortion, coping, stress, psychological wellbeing, complex intervention, mental health, randomized controlled trial

## Abstract

Background: Although undergoing an abortion is stressful for most women, little attention has been given to their psychological wellbeing. This protocol aims to assess the feasibility, acceptability, and primary effects of a complex intervention to promote positive coping behaviors and alleviate depression symptoms among Chinese women who have undergone an abortion. Methods: A two-arm randomized controlled trial design will be used. Participants will be recruited at their first appointment with the abortion clinic and randomly allocated to receive either the Stress-And-Coping suppoRT (START) intervention (in addition to standard abortion care) or standard care only. All participants will be followed-up at two- and six-weeks post-abortion. Approval has been granted by local and university ethics committees. This research was supported by an Australian Government Research Training Program Scholarship. Discussion: The results will assist refinement and further evaluations of the START intervention, contribute to improved abortion care practices in China, and enrich the evidence on improving women’s psychological well-being following abortion in China. Trial registration: Registered at the Chinese Clinical Trials.gov: ChiCTR2100046101. Date of registration: 4 May 2021.

## 1. Introduction

Induced abortion is common around the world. In 2015–2019, a third of all pregnancies ended in abortion (approximately 77.3 million abortions each year) [1]. Accessing good-quality abortion care is a fundamental human right, an important component of sexual and reproductive health services, and an essential way to achieve global health commitments [2] such as the Sustainable Development Goals (SDGs) [3]. Generally, abortion is physically safe when conducted using WHO-recommended methods by appropriately trained professionals [4]. For most women, the main challenge when choosing a first-trimester abortion is their emotional and psychological wellbeing [5]. Some women cope well with abortion and do not experience negative psychological consequences [6,7]. Some might even find positive meaning and embrace healthy behaviours after an abortion [8,9], whereas some report ongoing distress and mental health issues [5,10].

The Transactional Model of Stress and Coping helps to explain the psychological variability of abortion [10,11]. According to this model, abortion is a dynamic stressful life event that happens in response to an unwanted/unintended pregnancy. In this circumstance, a woman first appraises the significance/severity of the pregnancy and then evaluates her options and coping resources at her disposal [10,12]. A woman’s appraisal mediates her coping behaviours and the psychological consequences of an abortion (see Figure 1) [12]. The Transactional Model of Stress and Coping suggests that by offering women the support they need, they can adjust their coping processes and achieving positive health outcomes.

Despite the critical role of support to women accessing abortion, little attention has been paid to the psychosocial aspects of care [13]. A recently published systematic review found that in the past ten years, only ten experimental studies had been conducted to address the psychological needs of women undergoing abortion [14]. All interventions used a single-component design (e.g., music therapy, information support, or implementation of mandated waiting or counselling policies), and none were informed by a theoretical framework. Conflicting outcomes and methodological limitations hindered conclusions about which intervention could reasonably be adopted to improve women’s psychological well-being [14].

Abortion occurs in a socio-cultural context, and local policies regarding abortion accessibility play an important role in women’s psychological response toward an abortion. As the most populous country in the world, about 10 million abortions were conducted in China in 2019 [15]. China has one of the most liberal abortion policies amongst Asian countries, which allows first-trimester abortion without restriction, except for the prohibition of sex-selective abortions. Abortion is also socially acceptable in China, and most abortions that occur in China are due to unwanted pregnancies consistent with a societal desire for small families and timed birth. A recent study revealed that around a quarter of women seeking an abortion in China experienced high stress and moderate-to-severe depression [16]. Chinese women’s unmet support needs when seeking abortion were predictive of adverse psychological outcomes.

In the absence of available evidence, our research team developed the STress-And-coping suppoRT intervention (START). Underpinned by the Transactional Model of Stress and Coping, the START intervention will provide women with information, coping skills, and support to achieve positive short- and long-term health. The intervention is innovative and based on (1) current available evidence, (2) a sound understanding of the target population, (3) informed by Medical Research Council (MRC) guidance for complex interventions [17], (4) the Intervention Mapping (IM) framework [18], and (5) a five-step iterative pathway to gradually shape the intervention design [19].

The aim of this trail is to assess the START intervention among Chinese women undergoing an abortion. Its specific objectives are to:Determine the feasibility of the START intervention according to eligibility, recruitment, intervention delivery, and retention data;Evaluate the acceptability of the START intervention among participants based on the Theoretical Framework of Acceptability (TFA) of Health Care Interventions;Test the preliminary efficacy of the START intervention on women’s depression symptoms, coping behaviors, self-efficacy, perceived support levels, intimate relationship satisfaction, post-abortion personal growth, and abortion relevant outcomes compared with women receiving standard abortion care.

## 2. Materials and Methods

The study adopts a two-arm randomized controlled trial design with quantitative and qualitative information collected. Participants will be randomly allocated to receive either the START intervention in addition to standard abortion care or standard abortion care only with a 1:1 allocation rate. The study design is directed by the CONSORT statement and its extensions [20,21] and the Standard Protocol Items: Recommendations for Interventional Trials (SPIRIT) 2013 guideline for protocols of randomized trials [22]. The research activities inconsistent with the real-world clinical practice are outlined in Figure 2.

### 2.1. Setting

This study will be conducted in the Family Planning and Reproductive Health Centre (FPRHC) of a tertiary hospital in Beijing, China. Participants will be recruited from the outpatient unit of the FPRHC, where abortion-related services are normally provided.

### 2.2. Participants

Women will be invited to participate in the study if they meet the following criteria: (1) Chinese citizens or living in China; (2) can speak, read, and write in Chinese; (3) are 18 years or older; (4) seeking termination of an intrauterine pregnancy for non-medical reasons; (5) less than 14 gestational weeks; (6) certain about their abortion decision and indicate the decision is of their free will; and (7) own a smartphone. Women will be excluded if they: (1) are presenting to FPRHC for post-abortion follow-up examination or secondary treatment of an incomplete abortion); (2) require an abortion for medical reasons; (3) are currently receiving mental or psychological treatment; or (4) unable to give informed consent (e.g., severe intellectual disability).

### 2.3. Interventions

A psychologist as well as clinic staff who provide direct care will be responsible for maintaining the quality of care and women’s safety. See Table 1 for a detailed description of the START intervention and standard abortion care according to the template for intervention description and replication (TIDieR) [23]. Generally, the START intervention involves three interacting components: (1) a 30-minute face-to-face consultation immediately after a woman’s enrolment; (2) information support including a printed booklet and an online information platform available to them from the consultation until closure of the project; and (3) timely communication channels with health providers (a hotline staffed between the hours of 8 a.m. to 5 p.m. and a WeChat-based public profile page) valid until 6 weeks post-abortion. Table 2 presents the specific content of the information booklet and platform.

### 2.4. Outcomes

#### 2.4.1. Feasibility Outcomes

Feasibility outcomes include eligibility and intervention delivery data, which will be collect by a research logbook and participants’ completion of questionnaires. The specific outcomes are:Proportion of women who meet the eligibility criteriaProportion of eligible women who are recruitedProportion of recruited women who receive the allocated intervention to which they are randomizedWithdrawal or loss to follow-up rateMissing data rate

#### 2.4.2. Acceptability Outcomes

Acceptability refers to retrospective acceptability assessed after participating in the intervention. Phone calls will be made to all participants in the intervention group at the last follow-up. A questionnaire based on the TFA framework of Health Care Intervention [24] will guide the recorded phone interview. The questionnaire will include both closed- and open-ended questions representing all seven constructs of intervention acceptability: affective attitude, burden, perceived effectiveness, ethicality, intervention coherence, opportunity costs, and self-efficacy. Responses on closed-ended questions are on a Likert Scale from 1 to 5, and response labels vary depending on the item. For example, ‘to what extent was the intervention useful to you?’ has response options of 1 ‘not at all useful’ to 5 ‘extremely useful’. Responses to opened-ended questions will be noted and recorded. Examples of open-ended questions are ‘how do you feel about the intervention’, ‘which component of the intervention was most/least useful and why?’, and ‘did you encounter any barriers when participating in the intervention?’.

#### 2.4.3. Effectiveness Outcomes

Effectiveness outcomes include women’s depression symptoms (primary), coping behaviours, self-efficacy, perceived support levels, intimate relationship satisfaction, post-abortion personal growth, and abortion-relevant outcomes. Specific measures include:

##### Depression

Changes in women’s depression symptoms will be measured by the Patient Health Questionnaire (PHQ-9) [25]. The nine-item scale will be administered at the initial patient interview and monitor progress. Participants will be asked to rate the severity of their depressive symptoms on a four-point scale (0 = not at all; 1 = several days; 2 = more than half the days; and 3 = nearly every day). The nine items cover symptoms associated with experiencing pleasure, feeling down, and self-esteem. The total score can range from 0 to 27, with higher scores indicating greater severity of depression. The tool has been translated and validated with the Chinese population, showing strong psychometric properties (α = 0.86) [26].

##### Coping Behaviours

The Carver Brief COPE Inventory [27] will be used to measure changes in women’s coping behaviours. This 28-item inventory measures the use of 14 coping behaviours (2 items per strategy). Women are asked to rate how often they adopted each behaviour within the previous 2 weeks on a four-point Likert scale, ranging from 0 (I have not been doing this at all) to 3 (I have been doing this a lot). Coping behaviours are categorised into three dimensions: (1) problem-focused coping (four sub-scales: active coping, use of informational support, positive reframing, and planning); (2) emotion-focused coping (six sub-scales: emotional support, venting, humor, acceptance, religion, self-blame); and (3) dysfunctional-focused coping (four sub-scales: self-distraction, denial, substance use, and behavior disengagement). Brief COPE has been widely utilized in different populations, including women who experienced pregnancy termination for fetal abnormality [28]. The Chinese version of Brief COPE has shown adequate psychometric properties with a Cronbach’s alpha of <0.60 [29,30].

##### Self-Efficacy

The General Self-Efficacy scale (GSE-10) [31] has 10 items measuring optimistic self-beliefs to cope with difficulties in life. Items of GSE were scored on a four-point Likert scale, ranging from 1 (not at all true) to 4 (completely true). Scores range from 10–40 with higher total scores indicating that an individual feels more competent in dealing with stressful encounters. It is valid, reliable and has been translated into more than 30 languages, including Chinese [32]. The internal inconsistency of the Chinese version was 0.91 [32].

##### Perceived Social Support

The abbreviated version of the Medical Outcomes Study Social Support Scale (MOSSS-5) [33] and seven study-specific questions will be used to measure perceptions of support. The MOSSS-5 is a five-item reliable measure of perceived social support. Each item is scored on a five-point Likert scale and the items are summed for a total score ranging from 5−25. Its Chinese version has adequate test–retest reliability (0.89), and internal consistency items ask participants to rate their satisfaction, degree of happiness, feelings of reward, and comfort with their partners. Each question has different response formats. Total scores range from 0 to 21, with higher scores indicating greater satisfaction. The Cronbach’s α reliability of the translated version in Chinese women is 0.88 [34].

##### Post-Abortion Growth

The Post-traumatic Growth Inventory Short Form (PTGI-SF) evaluates personal growth following traumatic, challenging and stressful life circumstances [35]. PTGI-SF comprises 10 items rated on a six-point Likert scale (ranging from 0 to 5), with a total score ranging from 0 to 50. Higher scores indicated higher PTG. The 5 subscales include relating to others, new possibilities, personal strength, spiritual change, and appreciation of life. The Chinese version has acceptable internal consistency with a Cronbach α of 0.86 [36].

##### Abortion Related Outcomes

Abortion related outcomes will be measured according to women’s gestational weeks at abortion, type of abortion, complications, and follow-up-exam attendance.

### 2.5. Sample Size

A formal sample-size calculation is not necessary for research objectives 1 and 2 [20]. For objective 3, G*Power software was used to determine reasonable sample size for the estimation of differences between intervention and control groups. The effect size on depression reported by similar interventions ranged from 0.39 to 1.06 [37,38,39]. G*power indicated a total of 72 participants will be required to detect a 0.39 effect size for depression, two tailed, with alpha at 0.05 and power at 90%. Anticipating a 20% attrition rate, recruitment was set at 90 (45 women per group).

### 2.6. Recruitment and Consent

When women present at the FPRHC seeking an abortion, they receive a consultation to ensure informed choice and a medical appointment to assess their physical health. If the woman confirms her decision to terminate the pregnancy, she speaks with a nurse to make an appointment for the abortion. As reception nurses have an ongoing relationship with patients and are available during clinic hours, they will screen women based on the inclusion and exclusion criteria. If eligible, the reception nurse will provide a brief explanation about the study and ask women if they are interested in participating. If yes, women will be referred to the primary researcher, who will discuss the study in more depth, answer any questions, and provide the Participant Information and Consent Forms. The potential participant will be given adequate time to read the forms, talk with her support person, and ask questions before deciding to participate or decline. Participants will be assured they can withdraw from the study at any time, and it will not influence their treatment in any way.

### 2.7. Assignment and Blinding

Considering the small sample size, block randomization will be used to reduce bias and ensure an equal balance between the intervention and control groups [40]. A block size of four with six possible balanced combinations of assignment within the block is considered appropriate for this study. The nurse coordinator will generate a random number sequence using Microsoft Excel and place the assignments into sealed envelopes. After informed consent, the primary researcher will open a sequential envelope to determine the group allocation of each participant. Given the nature of the intervention, it is not feasible to blind the participants, intervention providers, and researchers who collect data. To prevent contamination, women in the intervention group will be asked not to share information or materials of the study with other abortion patients before completion of the study.

### 2.8. Data Collection

Data collection will occur at five timeframes: T-0, during recruitment before randomization; T-1, on abortion day before discharge (normally 2 h post-abortion procedure); T-2, during women’s post-abortion exam visit, normally 2 weeks post-abortion; T-3, 6 weeks post-abortion; and T-4, post completion of all follow-ups of all participants. See Table 3 for detailed information on data collection, including What, Who, When, Where, How, and by What data.

### 2.9. Data Analysis

Participant characteristics, feasibility outcomes, and quantitative acceptability outcome data will be summarized with descriptive statistics, including frequency counts and percentages (categorical variables) and mean with 95% confidence intervals (CIs) or standard deviation (continuous variables). Intervention feasibility and acceptability will be assessed against the following criteria: eligibility: ≥60% of patients screened will be eligible; recruitment: ≥80% of eligible participants will agree to participate; protocol fidelity: ≥90% of patients randomized to each group will receive the allocated intervention; retention: <20% of patients will be lost at 6 weeks after abortion; and missing data: <20% of data will be missing. Between-group differences of participant characteristics, baseline data, and effectiveness outcomes will be detected using Statistical Package for the Social Sciences (SPSS)^®^ version 23 [41]. Specifically, Chi square tests will be used for categorical/dichotomous variables, and *t*-tests or Mann–Whitney U tests for continuous variables (depending on normality). To capture changes in the effect outcomes over time, a general linear regression analysis will be performed. See Table 3 for the specific analysis approaches.

For qualitative data, audiotapes of telephone interviews will be transcribed and de-identified by the primary researcher. Transcripts and notes will be analyzed using the deductive approach of content analysis [42]. First, a structured analysis matrix will be developed based on the aims of the study and the TFA of Health Care Intervention. Then the primary researcher and nurse coordinator will independently review all de-identified data, and content fit the matrix. They will work together to group similar content within the matrix boundaries as categories, and then main categories. After abstraction of each category, the primary researcher and nurse coordinator will make a joint decision on how to model and report the results. Any discrepancies along the process will be discussed with a third reviewer until an agreement is reached. To preserve the original meaning of women’s statements, the data analysis process will be conducted using Chinese. Development of the analysis matrix and reporting process will use Chinese and English to reduce potential bias and build a shared vision among the research team.

### 2.10. Protocol Modifications

During the conduction of the trial, if we observe a higher-than-expected drop-out rate at the two-weeks post-abortion follow-up, we will increase the sample size from 90 (calculated on a 20% attrition rate) to 110 (calculated on a 35% attrition rate) to achieve precision for the estimation of the effect size. Paper-based questionnaires will be left for women to fill out on their own. However, if participants express being overwhelmed by the amount of information required, we will conduct an interview during which participants could ask clarifying questions. This approach will minimize missing data and enhance the retention rate.

## 3. Discussion

We propose a two-armed randomized trial among Chinese women undergoing abortion, to assess the feasibility, acceptability, and potential effects of the START intervention. To our knowledge, this study will be the first trial to address Chinese abortion patients’ psychological wellbeing by providing a comprehensive supportive intervention based on the Transactional Model of Stress and Coping [19]. This innovative intervention contains different active components that will be delivered using different methods. Findings from this trial will allow us to evaluate the feasibility and acceptability of the intervention, which will inform intervention refinement [17]. The findings will also help to identify potential barriers and facilitators for implementing the intervention in the Chinese context [43]. As the preliminary effectiveness and effect of the intervention will also be described, the study could have important implications for relevant policymakers, practitioners, researchers, or other stakeholders [44].

We are employing a rigorous study design to ensure the internal and external validity of findings. This trial responds to the call for well-designed, rigorously conducted scientific research in abortion care in developing countries [1]. Randomization and statistical analysis adjusted for any difference in baseline characteristics between groups will protect against selection bias, and ultimately warrant internal validity [45]. The study will be performed under real-world clinical practice conditions, which will enhance the external validity and generalizability of results. All instruments used in this study have been validated in the Chinese population, which will also enhance the reliability of findings.

A further strength is that we have adopted a series of appropriate guidelines to ensure the normalization, transparency, and replication of the study. The TIDieR checklist has been used to describe the intervention [23]. Application of the CONSORT statements [20,21] and the SPIRIT 2013 Checklist [22] have been followed.

Limitations, however, need to be considered. The emotional period between a woman’s initial appointment with the clinic and 6 weeks post-abortion [46], as well as the sensitive nature of the topic, could challenge women’s adherence and completion of the study. Strategies such as keeping information simple, offering the intervention in confidential formats, and repeated reminders that are known to be effective in improving intervention adherence will be employed [47]. Another challenge lies in the uncertainties of the COVID-19 pandemic. We will follow and document all local, national, and international measures that may impact our study progress. The actual delivery of the intervention and identified barriers and facilitators of implementing the intervention will be well-recorded to inform future studies.

## 4. Conclusions

This paper proposes a protocol of a clinical trial to assess the feasibility, acceptability, and potential effects of a comprehensive supportive intervention that was developed to address the psychological health of Chinese women undergoing an abortion. Results will be used to assist intervention refinement and direct a future full-powered trial. Practical experience will be gained throughout the clinical trial, which would also provide evidence for further implementation of the intervention.

## Figures and Tables

**Figure 1 ijerph-19-06611-f001:**
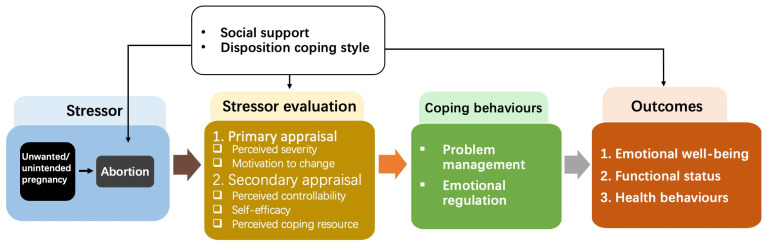
The Transactional Model of Stress and Coping.

**Figure 2 ijerph-19-06611-f002:**
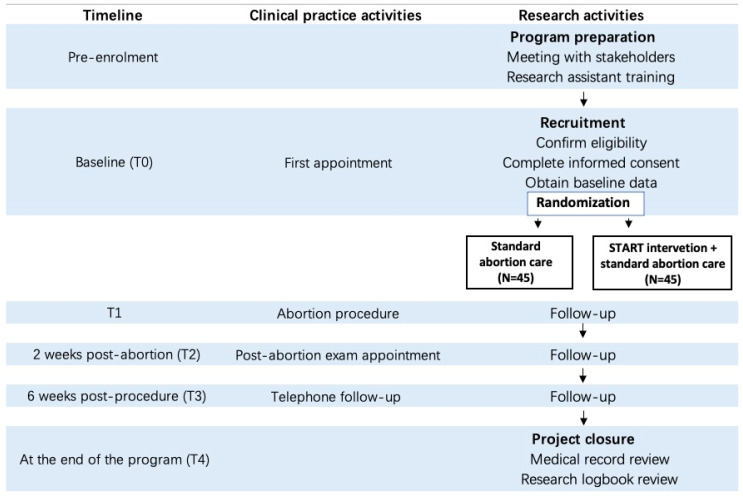
SPIRIT standard flow diagram.

**Table 1 ijerph-19-06611-t001:** Intervention description according to TIDieR checklist.

TIDieR Item	Intervention Group	Control Group
**1.** **BRIEF NAME**	The START intervention	Standard abortion care
**2.** **WHY**	Based on the Transactional Model of Stress and Coping, the START intervention intends to help women cope with abortion experience by addressing the specific stressors they face in the Chinese context, hence promoting well-being	N/A
**3.** **WHAT-Materials**	Consultation, an information booklet, a WeChat-based public profile page, and a hotline	N/A
**4.** **WHAT-Procedure**	All participants allocated to the intervention group will be offered a 30-min face-to-face consultation. The consultation includes a brief introduction of the public profile page, Q&A, and offering emotional support and information on available coping resources. A printed information booklet will be provided during the consultation. Meanwhile, by scan the QR code, participants will be endowed with unlimited access to the public profile page till the closure of the project. A hotline will also be offered.	(1) prior abortion consultation on informed choice of pregnancy; (2) abortion preparation education on abortion practice, and steps involved; (3) a brief discharge education session on warning signs of potential complications and post-abortion contraception; and (4) a regular 2-week post-abortion appointment to determine the completion of the abortion.
**5.** **WHO PROVIDED**	The consultation, information booklet, and managing the public profile page will be carry out by the primary researcher who is a qualified abortion counsellor. Hotline will be answered by a registered nurse who has more than 10 years clinical experience in abortion care.	Prior abortion consultation and post-abortion exam-by doctors Abortion preparation and discharge education-by nurses
**6.** **HOW**	Consultation (face to face) Information booklet (hard copy) Public profile page (internet) Hotline (telephone)	Face to face
**7.** **WHERE**	Consultation (the consultation room) Information booklet (the consultation room) Public profile page (WeChat based) Hotline (telephone-base)	Health centre facilities
**8.** **WHEN and HOW MUCH**	Consultation (after women made their abortion appointment, 30 min) Information booklet (during the consultation, one copy to take home) Public profile page (from the consultation, unlimited access until the closure of the project) Hotline (from the consultation, available during workhours)	Prior abortion consultation—at women’s first appointment (10–20 min); Abortion preparation education—after women made their abortion appointment (5–10 min); post-abortion education—before women are discharged (10 min); The post-abortion exam—2 weeks after abortion
**9.** **TAILORING**	Participants in the intervention group will receive the same intervention	N/A
**10.** **MODIFI-CATION**	Cannot be reported until the study is complete	N/A
**11.** **HOW WELL**	The research nurse and reception nurses involved in the trial will be trained initially. A researcher logbook will be kept recording intervention delivery of each participant. Adherence to the intervention will be assessed by participants self-reported usage info collected during follow-ups	N/A
**12.** **ACTUAL**	Cannot be reported until the study is complete	N/A

Abbreviations: Q&A: questions and answers; N/A: not applicable.

**Table 2 ijerph-19-06611-t002:** Content of the information booklet and platform.

Categories	Themes	Example Items
Abortion	Facts about abortion	General information about abortion Pregnancy calculation and ectopic pregnancy Abortion practice in Beijing
	Different types of abortion	Medical abortion and steps involved Aspiration/surgical abortion and steps involved Features of aspiration abortion and medical abortion
	Q&A	What is your legal position, e.g., gestation age limits; no need for husbands’ or partners’ permission; and Nation ID required etc. How much does abortion cost? Does abortion hurt and what is a painless induced abortion? What will you experience during an abortion? How could you know your abortion is complete? Will an abortion cause infertility? How will you feel if you have an abortion
	Aftercare and follow-up examination	Follow-up examination and why it is necessary Symptoms requiring contacting your health provider, going to local emergency room, or calling 120 for Emergency Health Aid.
Intimate Relationship	A healthy relationship	Important characteristics of a healthy relationship Rights and responsibilities in a relationship Equality in intimate relationships
	Identify and deal with a dangerous relationship	Signals of unhealthy relationships Options for women experiencing an unhealthy/dangerous relationship Recognising Intimate Partner Violence (IPV) Options for women experiencing IPV
	About sex	Consent and your legal position Options for women who are forced to engage in a sexual activity
Available Coping Resources	Coping with abortion experience	Suggestions from experts (Chinese Health & Medical Development Foundation) Suggestions from women who have experienced an abortion
	Taking care your emotions	Talk to someone you trust: Tips on finding a reliable person to talk to (Marie Stopes China) Medication: free medication; Apps recommendation e.g., Tide Reading: e.g., Trauma and Recovery by Judith Herman _Chinese version Post-abortion counsellor
	Cope with IPV or sexual violence experience	Free social support services e.g., Beijing Women’s Federation hotline; The Maple Women’s Psychological Counselling Center hotlines etc. Self-help guides e.g., Guidelines for dealing with sexual assault (Marie Stopes China); SARSAS Suggestions from police Stories from IPV survivors

Abbreviations: ID: identification; IPV: intimate partner violence; Q&A: questions and answers; SARSAS: Self-help guide to rape and sexual abuse—Chinese version.

**Table 3 ijerph-19-06611-t003:** Data collection schedule and analysis method.

Concept (What)	Time (When)	Method (Who, Where, and How)	Measure (by What)	Method of Analysis
**Participant characteristics**	T-0	Primary researcher, at the clinic, face to face	Self-design questionnaire on sociodemographic characteristics, health history, and current gestation	Descriptive statistics X^2^ test comparing differences between groups of categorical/dichotomous variables *t*-test/Mann–Whitney U test comparing differences between groups of continuous variables
**Feasibility outcomes**				
Eligibility	T-0	Reception nurses, at the clinic, face to face	Eligibility screening checklist	Descriptive statistics
Intervention delivery	T-4	Primary researcher, at clinic facilities, researcher logbook review	Feasibility indexes include intervention fidelity, retention rate, and missing data rate	Descriptive statistics
**Acceptability outcomes**				
Acceptability	T-3	Primary researcher; at clinic facilities, telephone-interview	Self-design questionnaire including closed-ended and open-ended questions	**Quantitative data** Descriptive statistics Chi Square test comparing differences between groups of categorical variables **Qualitative data** A deductive approach of content analysis
**Effectiveness outcomes**				
Depression	T-0, -1, -2, -3	Primary researcher (T0, 3) and nurse coordinator (T-1, -2); at clinic facilities; face to face (T-0, -1, -2) and by telephone (T-3)	SMD-A	Descriptive statistics *t*-test/Mann–Whitney U test comparing baseline to T-3 *t*-test/Mann–Whitney U test comparing differences between groups at T-3 Linear regression measuring change over time
Perceived support	T-0, -1, -2, -3	As above	MOS-SSS	As above
Intimate relationship	T-0, -1, -2, -3	As above	CSI-4	As above
Coping behaviours	T-0, -2, -3	As above	CBCI	Descriptive statistics *t*-test/Mann–Whitney U test comparing baseline to T-3 of each coping style *t*-test/Mann–Whitney U test comparing differences between groups of each coping style at T-2 and T-3
Self-efficacy	T-0, -2, -3	As above	GSE-10	Descriptive statistics *t*-test/Mann–Whitney U test comparing baseline to T-3 *t*-test/Mann–Whitney U test comparing differences between groups at T-2 and T-3
Post-abortion growth	T-2, -3	As above	PTGI-SF	Descriptive statistics *t*-test/Mann–Whitney U test comparing T-2 to T-3 *t*-test/Mann–Whitney U test comparing differences between groups at T-2 and T-3
Abortion outcomes	T-3	As above	Self-design questions	Descriptive statistics Chi Square test comparing differences between groups at T-3

## Data Availability

The data involved in this study will be provided upon reasonable request to the corresponding author.

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
