# Peer review of "The STress-And-Coping suppoRT Intervention (START) for Chinese Women Undergoing Abortion: A Randomized Controlled Trial Protocol"

_ijerph, 2022, doi:10.3390/ijerph19116611_

Round 1

Reviewer 1 Report

The area of investigation is important. A simple intervention for women undergoing abortion is much needed. As this is a protocol paper, the cultural components affecting women undergoing abortion should be elaborated. The cause or reason leading to their decision of abortion should also be collected. 

In the intervention booklet, there is lots of mention of IPV, forced sex etc. Is this unique to China? I suppose the sample hospital is in a major city in China. What proportion of these women have children already? Proportion of women being victim of rape and sexual violence by non-partners? Is it difficult to get an abortion when out of marriage? The China situation in contemporary society and basic statistics should be elaborated. 

In the description of intervention: "three interacting components: 1) a 30-minute face-to-face consultation immediately after a woman’s enrollment; 2) information support including a printed booklet and an online information plat-form available to them from the consultation until closure of the project; and 3) timely communication channels with health providers valid until 6 weeks post-abortion." It is unclear how the "three" interacting components is being carried out? Is it a 5 minute, 10 minute, 15 minute face-to-face consultation or phone intervention?  Details of "timely communication channels with health providers valid until 6 weeks post-abortion" should be provided. What exactly is timely communication?

The use of English needs careful editing. For example, should "sexually activity" be sex activity instead?

Reviewer 2 Report

Dear author

The article presented " The STress-And-coping suppoRT intervention (START) for 2 Chinese women undergoing abortion: A randomized controlled trial protocol ", is interesting because publishing protocols of randomized controlled trials facilitates a more detailed description of the study rationale, design, and related ethical and safety issues, which should promote transparency.

The problem addressed in the study protocol that they intend to implement seems to me current and relevant. This is because there has been a concern in recent years that abortion itself may increase psychological risk and adversely affect the woman's mental health. On the other hand, the last two years of the pandemic have given visibility to the importance of positive mental health throughout the life cycle, with particular importance in the reproductive cycle.

The protocol of the study was done correctly, and the methodology adequately evaluates the proposed objectives. However, it should be a little more detailed in the description of the instruments.

As long as we can evaluate as a non-native English speaker, the language is adequate and correct.

We recommend authors improve certain aspects to increase the manuscript's clarity. Several comments are below.

Abstract

Line 13: Although the background in the abstract should be brief, I suggest that it could be improved;

Line 13: Authors should mention the purpose of publishing the protocol in addition to the objectives of the study;

Line 18: Clarify the meaning of the acronym START;

Line 21: The acronym RTP is not necessary for the abstract.

Introduction

Line 30: It would be important to introduce prevalence data on abortion induction in the Chinese population because there are important variations between different countries and regions. As well as prevalence data of stress and depression and associated factors among women seeking an induced abortion.

Line 62 to 82: The information about the objectives of the study seems to be duplicated, so it should be revised to be clearer and more consistent in the abstract and in the introduction.

Authors must include the publication objectives of the protocol.

Materials and Methods

In this subchapter, in general, the description of the different scales must be standardized, mentioning the objectives, measures, denomination of the dimensions of the scales (if applicable), total scores and evaluation sense, and psychometric validation data of the scales.

Line 88 - Clarify the meaning of the acronym SPIRIT.

Line 90- In Figure 2. correct the nonconformities of the text and formatting.

Line 151 to 159 – Depression: Include the psychometric validation data of this instrument.

Line 160 to 170 – Coping behaviours: Authors should clarify how coping behaviours are assessed.

Authors should clarify how coping strategies and differences in the revised and the original scale are evaluated.

Line 171 - Self-Efficacy: Describe the scoring and reading measures of the scale's global scores.

Line 187- Intimate relationship: The description of this scale can be more detailed.

Line 176- Perceived Social Support: What is the meaning of scale evaluation?

References

The references are adequate, although some are old. I understand that they mainly concern documents that support the methodological part of the research protocol.

Reviewer 3 Report

The manuscript under review is a research project that will assess whether the START procedure will improve psychological care for Chinese women undergoing abortions.

The methodology of the presented protocol is debatable. The authors revealed that there will be two groups of 45 women. The recruited patients will be assessed within 6 weeks after the abortion. Will the course of this study be supervised by psychologists or only by midwives and nurses?

The most important issue is the reason for the “non-medical” abortion - given the specificities of these abortions in China - ethical issues appear to be of key importance for this protocol.

In my opinion, the purpose of this publication does not seem fully understandable. It would make more sense to present the results of this study, and not just describe what the authors plan to carry out.

Round 2

Reviewer 3 Report

The authors have appropriately addressed my concerns and suggestions in their revised version.